# Construction of a Novel Multigene Panel Potently Predicting Poor Prognosis in Patients with Clear Cell Renal Cell Carcinoma

**DOI:** 10.3390/cancers12113471

**Published:** 2020-11-22

**Authors:** Xiaozeng Lin, Anil Kapoor, Yan Gu, Mathilda Jing Chow, Jingyi Peng, Pierre Major, Damu Tang

**Affiliations:** 1Department of Medicine, McMaster University, Hamilton, ON L8S 4L8, Canada; linx36@mcmaster.ca (X.L.); guy@mcmaster.ca (Y.G.); zhouj32@mcmaster.ca (M.J.C.); pengj31@mcmaster.ca (J.P.); 2Urological Cancer Center for Research and Innovation (UCCRI), St Joseph’s Hospital, Hamilton, ON L8N 4A6, Canada; akapoor@mcmaster.ca; 3The Research Institute of St Joe’s Hamilton, St Joseph’s Hospital, Hamilton, ON L8N 4A6, Canada; 4Department of Surgery, McMaster University, Hamilton, ON L8S 4L8, Canada; 5Department of Oncology, McMaster University, Hamilton, ON L8S 4L8, Canada; majorp@hhsc.ca

**Keywords:** clear cell renal cell carcinoma, IQGAP1, overall survival, disease-specific survival, progression-free survival, metastasis

## Abstract

**Simple Summary:**

Clear cell renal cell carcinoma (ccRCC) is the predominant cause of kidney cancer death attributed to its prevalence (70%) and its nature being the most aggressive form of kidney cancer. Most ccRCC deaths are resulted from metastasis. It is essential to know which ccRCCs are at risk of metastasis and the development to lethal disease; however, our capacity for such analysis remains poor. To improve this diagnostic capacity, we have examined a comprehensive ccRCC dataset containing 512 patients and have produced a 9-gene signature. This signature is novel; all its 9 components genes are unknown to be related to ccRCC. Importantly, all 9 individual genes possess significant ability in diagnosis of ccRCC metastasis and fatality; the combination of these genes or this signature predicts deadly ccRCCs at an impressive efficiency. This research will open new avenues in ccRCC research and will have a major impact in reducing ccRCC-associated deaths.

**Abstract:**

We observed associations of IQGAP1 downregulation with poor overall survival (OS) in clear cell renal cell carcinoma (ccRCC). Differentially expressed genes (DEGs, *n* = 611) were derived from ccRCCs with (*n* = 111) and without IQGAP1 (*n* = 397) reduction using the TCGA PanCancer Atlas ccRCC dataset. These DEGs exhibit downregulations of immune response and upregulations of DNA damage repair pathways. Through randomization of the TCGA dataset into a training and testing subpopulation, a 9-gene panel (SigIQGAP1NW) was derived; it predicts poor OS in training, testing, and the full population at a hazard ratio (HR) 2.718, *p* < 2 × 10^−16^, *p* = 1.08 × 10^−5^, and *p* < 2 × 10^−16^, respectively. SigIQGAP1NW independently associates with poor OS (HR 1.80, *p* = 2.85 × 10^−6^) after adjusting for a set of clinical features, and it discriminates ccRCC mortality at time-dependent AUC values of 70% at 13.8 months, 69%/31M, 69%/49M, and 75.3%/71M. All nine component genes of SigIQGAP1NW are novel to ccRCC. The inclusion of RECQL4 (a DNA helicase) in SigIQGAP1NW agrees with IQGAP1 DEGs enhancing DNA repair. THSD7A affects kidney function; its presence in SigIQGAP1NW is consistent with our observed THSD7A downregulation in ccRCC (*n* = 523) compared to non-tumor kidney tissues (*n* = 100). Collectively, we report a novel multigene panel that robustly predicts poor OS in ccRCC.

## 1. Introduction

Kidney cancer ranks the 6th and 8th most common cancer in men and women, respectively, in USA [1]; the disease is the 9th and 14th most common cancer in men and women, respectively, worldwide (WHO Cancer Report 2020; https://www.wcrf.org/dietandcancer/cancer-trends/kidney-cancer-statistics) [2]. Renal cell carcinoma (RCC) accounts for 85% of kidney cancer cases; the most common subtypes are clear cell RCC (ccRCC, 80%), papillary RCC (pRCC, 15%), and chromophobe RCC (5%) [3]. Clear cell RCC is the most aggressive RCC and contributes to the majority of kidney cancer deaths. The main curative treatment for primary ccRCC remains complete and partial nephrectomy; in these patients, 20–40% will eventually experience recurrence [4,5] with a metastatic rate of 5–15% [5]. Metastatic ccRCCs are currently treated with immunotherapies aiming to block immune checkpoints and targeting vascular endothelial growth factor (VEGF) using tyrosine kinase inhibitors (TKIs) [6]. The first-line therapies involve checkpoint inhibitors (anti-CTLA-4 plus anti-PD-L1) [7] or a combination of either anti-PD-1 or anti-PD-L1 with the TKI axitinib [8,9]. Inhibitors of mTOR could be used in second-line therapy [6]. Targeting VEGF is developed based on the long-term investigations demonstrating loss of the von Hippel-Lindau (VHL) tumor-suppressor, due to 3p loss [10], mutations [11], and promoter methylation [12], as the common initiating event of ccRCC [13,14]. Loss of VHL leads to stabilization of hypoxia-inducible factors (HIF), which upregulates VEGF. In addition to the loss of VHL, other oncogenic events are required for ccRCC; one being the activation of the phosphoinositide 3-kinase (PI3K)-AKT-mTOR pathway [13,14]. The current therapies via targeting immune checkpoints, VEGF, and mTOR clearly improved overall survival (OS) and progression-free survival (PFS) in patients with metastatic ccRCC [6,7,8,9]. However, even with the recently formulated first-line therapies involving combinations of immune checkpoints inhibitors and TKI, the development of therapeutic resistance in metastatic ccRCCs remains common [6]. One way to improve patient management is to effectively classify primary ccRCCs with a high risk of recurrence, metastasis, and death. Currently, TNM staging and other clinical models are used to provide the prognostic assessment. However, this system is far from being perfect; molecular biomarkers are required to improve the assessment of ccRCC progression.

A variety of serum and urine biomarkers have been actively investigated, including circulating tumor cells, cell-free tumor DNA, circulating RNA, and proteins. These biomarkers need substantial validation [4]. Most recently, the field of ccRCC biomarker has been systemically evaluated and updated [15]. Among the 28 biomarkers examined, ccB was the only independent prognostic biomarker after adjusting for tumor grade and stage [15]. Effective ccRCC biomarkers need to be further developed.

IQGAP1 belongs to the IQ motif GTPase-activating scaffold proteins (IQGAPs). IQGAP1 facilitates ERK activation, binds Cdc42 and Rac1, and stabilizes their GTP association, thereby induces cytoskeleton reorganization [16]. With these activities, IQGAP1 promotes tumorigenesis and is upregulated in numerous cancer types [17]. On the other hand, IQGAP1 shares an overall 62% homology with IQGAP2; the two proteins have even higher similarities between their respective structural motifs, except for the WW domain [16,17]. Surprisingly, IQGAP2 is a tumor-suppressor [17,18]. Of note, downregulation of IQGAP1 has been reported to associate with tumor progression and poor prognosis in bladder cancer [19]; IQGAP1 suppresses TGFβ-mediated myofibroblast activation in tumor stroma, and thus, inhibits tumor metastasis in the liver [20], revealing IQGAP1 possession of tumor-suppression activities. Nonetheless, whether IQGAP1 is involved in ccRCC remains to be investigated.

We performed a systemic study on the potential biomarker values of IQGAP1 in ccRCC. Using the TCGA PanCancer Atlas ccRCC dataset (*n* = 512), which can be used for survival analysis [21], within cBioPortal [22,23], we observed a significant association of IQGAP1 downregulation with OS shortening. Differentially expressed genes (DEGs, *n* = 611) have been identified relative to IQGAP1 downregulation. Using the Elastic-net regression with 10-fold cross-validation, a 9-gene signature was constructed from a training population obtained by random division of the PanCancer cohort. This signature is novel; among the nine genes, three have not been reported for oncogenic involvement, and none has been reported in ccRCC. Importantly, this signature and its component genes robustly associate with reductions in OS, disease-specific survival (DSS), progression-free survival (PFS), and the adverse features (high grade, advance stage, and metastasis) of ccRCC. Collectively, this research has generated a novel and robust gene signature with promising clinical applications in managing ccRCC patients.

## 2. Results

### 2.1. Association of IQGAP1 Downregulation with Decreases of OS in ccRCC Likely via Impacts on Multiple Processes

We have previously demonstrated IQGAP2 as a potential tumor-suppressor [18] and noticed a high level of conservation between IQGAP1 and IQGAP2 [16,17]. IQGAP1 promotes tumorigenesis in multiple cancer types [17], but its role in ccRCC remains unclear. By using the GEPIA2 database (http://gepia2.cancer-pku.cn/#index) [24], we showed an association of IQGAP1 downregulation with shortening of overall survival (OS) in ccRCC at a hazard ratio (HR) 0.68 (*p* < 0.05) (Appendix A). We further observed that reductions of IQGAP1 mRNA expression at −0.6 SD (standard deviation or z-scores at −0.6), −0.8 SD, and −1 SD significantly stratified ccRCCs into high- and low-risk groups of fatality within the TCGA PanCancer Atlas ccRCC dataset (Appendix A). To further analyze this association, we obtained differentially expressed genes (DEGs) relative to IQGAP1 downregulation. Based on the separating ccRCCs into high and low fatality risk groups at the cutoff points of −0.6 SD, −0.8 SD, and −1 SD (Appendix A), we divided the TCGA PanCancer Atlas ccRCC cohort into a high-risk group with IQGAP1 downregulation (*n* = 111) and a low-risk group without the reduction (*n* = 399) using −0.8 SD cutoff point. From these two groups, a set of DEGs were obtained; at q < 0.0001, *n* = 6563 DEGs were identified. As IQGAP1 is reduced at the log2 value of −0.95 (fold 1.93 reduction) in the high-risk group (q = 2.54 × 10^−57^), we have thus defined DEGs at q < 0.0001 and fold change ≥ |1.93| or log2 value ≥ |0.95|, which resulted in *n* = 611 genes (Appendix A) that are differentially expressed in ccRCCs with concurrent IQGAP1 downregulation (high-risk group) compared to those without the downregulation (low-risk group).

To analyze the pathway or processes affected by the IQGAP1 network (DEGs), enrichment analysis was performed using the Metascape network (https://metascape.org/gp/index.html#/main/step1) [25]. A set of top enriched non-redundant clusters GO (gone oncology) terms of biological processes (BP), and KEGG pathways were obtained; the top representative GO BP terms and KEGG pathways of the top 20 clusters are shown (Figure 1A), and the networks of these enriched clusters are included (Figure 1B). The inclusion of the enriched cluster of positive regulation of cellular component movement, chemotaxis, focal adhesion, response to mechanical stimulus, regulation of cell adhesion, extracellular extravasation, and cell matrix adhesion (Figure 1A) is in line with the classical knowledge of IQGAPs being involved in cytoskeleton organization [16]. Other critical oncogenic processes affected by IQGAP1 DEGs include angiogenesis, response to PI3K signaling, and immune responses (including cytokine biosynthetic process and macrophage activation) (Figure 1A). Collectively, evidence indicates alterations of multiple critical oncogenic processes in ccRCCs with downregulated IQGAP1 expression.

The above concept is further supported by gene set enrichment analyses. A robust downregulation of immune responses is observed, including downregulations of interferon γ response, IL-2-STAT5 signaling, allograft rejection, TNFα signaling, complement, inflammatory response, and others (Figure 2, Table 1). On the other hand, oxidative phosphorylation and DNA damage repair are enriched (Figure 2, Table 1). The enrichment of the former is in accordance with the alteration of the oxidative phosphorylation cluster derived from Metascape-based enrichment analyses (Figure 1A). Taken together, we provide a comprehensive analysis of alterations in multiple oncogenic processes that are associated with IQGAP1 reductions in ccRCC.

### 2.2. Construction of SigIQGAP1NW to Predict OS in ccRCC Following Nephrectomy

To further examine the oncogenic impact of IQGAP1 DEGs in ccRCC, we determined the network’s potential to predict OS. This research effort is supported by two observations. (1) The TCGA PanCancer ccRCC cohort contains comprehensive genetic and gene expression data obtained from primary or localized ccRCCs following resection; the cohort has been curated to support survival outcome analyses, including OS, disease-specific survival (DSS), and progression-free survival (PFS) [21]. (2) As all ccRCCs of the cohort were primary or organ-confined tumors [21], the pathways or gene sets enriched in the IQGAP1 DEG network suggest the network playing important roles in ccRCC progression, a process relevant to ccRCC OS. We randomly divided the TCGA PanCancer ccRCC cohort into a training (*n* = 300) and testing (*n* = 208) population at the ratio of 6:4. The effectiveness of randomization was confirmed based on the distributions of age and other clinical features (Appendix A). From the training sub-population, we performed six rounds of covariate selection among the 611 DEGS (Appendix A) for impact on OS using Elastic-net within the R *glmnet* package with the mixing parameter α set at 0.5 and 10-fold cross-validation. All unique DEGs identified by these selections constitute the final multigene panel (SigIQGAP1NW with NW representing network), which includes nine genes (Table 2).

We first confirmed the effectiveness of SigIQGAP1NW in the evaluation of mortality risk in the training group. SigIQGAP1NW scores for individual ccRCCs were calculated as ∑(f_i_)_n_ (f_i_: Cox coefficient (coef) of gene_i_ × gene_i_ expression, *n* = 9). Individual Cox coefs were obtained by multivariate Cox analysis. The scores significantly predict the fatality risk at HR = 2.72, 95% CI = 2.16–3.44, and *p* < 2 × 10^−16^ (Figure 3A). Increases in SigIQGAP1NW score are also associated with poor disease-specific survival (DSS) and shortening of progression-free survival (PFS) (Figure 3A). SigIQGAP1NW scores discriminate ccRCC fatality with time-dependent area under curve (tAUC) values ranging from 71.8% at 14.6 months (71.8%/14.6 M) to 80.2%/72.6 months (Figure 3B). The tAUC values for DSS and PFS are 76.9%/14.7 M to 81%/62.6 M and 62.2%/8.1 M to 70.7%/64.1 M, respectively (Figure 3B). With the respective cutoff points for OS, DSS, and PFS defined by Maximally Selected Rank Statistics (Appendix A), SigIQGAP1NW robustly classifies high- and low-risk groups based on OS, DSS, and PFS, respectively (Figure 3C).

### 2.3. Testing SigIQGAP1NW

To mimic clinical diagnosis applications, we first validated SigIQGAP1NW biomarker potential to predict OS in the testing group using those coefs produced from the training group, i.e., SigIQGAP1NW scores based on the setting of the training cohort. In the testing group, SigIQGAP1NW stratifies fatality risk at HR = 2, 95% CI = 1.31–3.06, and *p* = 0.00137 and effectively separates ccRCCs in the testing cohort into a low- and high-risk mortality group (Figure 4A). To reveal the full potential of SigIQGAP1NW in the evaluation of fatality risk, we rederived component gene coefs and recalculated SigIQGAP1NW scores based on the testing population following the system described above. SigIQGAP1NW potential can be significantly enhanced based on HR and its enhanced efficiency in separating the low- and high-risk group (Figure 4B, see its comparison with panel A). Impressively, the prediction rate in the high-risk group reaches 74.1% (20/27), and patients in this group showed a substantially reduced survival time (Figure 4B). The efficiency in the prediction of OS, DSS, and PFS has the respective tAUC values of 70.6%/13M, 62.1%/12.8M, and 68%/7.9M (Figure 4C). Collectively, SigIQGAP1NW effectively predicts OS, DSS, and PFS in the testing population, which significantly enhances the biomarker value of SigIQGAP1NW.

Similar observations were also obtained on the full TCGA PanCancer Atlas ccRCC cohort. The training group-produced SigIQGAP1NW scores predict poor OS at HR = 2.04, 95% CI = 1.56–2.63, *p* = 3.11 × 10^−8^. When using the full cohort-derived signature scores, SigIQGAP1NW robustly stratifies patients with a high risk of mortality from those with a low-risk (Figure 5A). Along with predicting OS, SigIQGAP1NW efficiently predicts the risk of DSS and PFS (Figure 5B,C).

### 2.4. Association of SigIQGAP1NW with Worse Clinical Features of ccRCC

To further examine SigIQGAP1NW biomarker potential, we investigated its relationship in predicting the fatality risk of ccRCC with other established clinical features. As expected, age at diagnosis, stage, tumor grade, and Winter Hypoxia Score are all clear risk factors of poor OS (Table 3). Activation of hypoxia is a major oncogenic driver in ccRCC owing to the common inactivation of the tumor-suppressor VHL [13,14]. The level of hypoxia can be quantified based on a 99-gene hypoxia signature defined by Winter et al. [26]. Of note, increases in Winter Hypoxia Score predict poor OS. After adjusting these clinical factors along with Winter Hypoxia Score, increases in SigIQGAP1NW score remain significant in its prediction of fatality at HR = 1.80, 95% CI = 1.41–2.31, *p* = 2.85 × 10^−6^ in the TCGA PanCancer Atlas ccRCC cohort (Table 3). Consistent with these observations, SigIQGAP1NW is significantly associated with worse clinical features of ccRCC, including high tumor grade and overall stage, including T stage and distant metastasis (M stage) (Figure 6). Since all ccRCCs included in TCGA PanCancer cohort were primary tumors (see the clinical data in cBioPortal) [21], the association with distant metastasis (Figure 6) supports a potential of SigIQGAP1NW to predict the risk of ccRCC metastasis. Additionally, SigIQGAP1NW score predicts poor OS in M0 ccRCCs at HR = 3.11, 95% CI = 2.31–4.17, and *p* = 3.65 × 10^−14^, as well as primary ccRCCs with confirmed distant metastasis (the “M1” ccRCCs) at HR = 1.82, 95% CI = 1.31–2.57, and *p* = 0.000421. Collectively, evidence supports SigIQGAP1 in assessing the risk of primary ccRCCs progression to a metastasis disease. Furthermore, all nine component genes of SigIQGAP1NW are individually associated with OS shortening (Table 4). Except THSD7A, LINC01089 (*p* = 6.57 × 10^−9^), SPACA6 (*p* = 4.18 × 10^−6^), LOC155060 (*p* = 2.62 × 10^−5^), LOC100128288 (*p* = 6.82 × 10^−6^), SNHG10 (*p* = 2.76 × 10^−6^), RECQL4 (*p* = 0.00661), HERC2P2 (*p* = 4.15 × 10^−6^), and ATXN7L2 (*p* = 4.01 × 10^−5^) are all independent risk factors of poor OS following the adjustment for age at diagnosis, sex, stage, tumor grade, and Winter Hypoxia Score. Taken together, evidence supports SigIQGAP1NW being a novel and robust multigene panel in predicting OS of ccRCC.

### 2.5. SigIQGAP1NW, a Novel Multigene Panel of ccRCC Biomarker

In view of all nine component genes being individual predictors of OS shortening (Table 4), we have analyzed their oncogenic potential. It was noticed that the directionality of these genes in predicting poor OS are in accordance with their differential expression in ccRCCs with IQGAP1 downregulation. For instance, THSD7A is co-downregulated with IQGAP1 (Table 2), and its expression levels are reversely associated with poor OS (HR < 1; Table 4), which is in line with IQGAP1’s relationship with the fatality risk of ccRCC (Appendix A). On the other hand, the rest of the component genes are upregulated in ccRCC with IQGAP1 downregulation (Appendix A), and elevations in their expressions predict poor OS (HR > 1; Table 4). These observations further support the association between IQGAP1 downregulation with increases in mortality risk of ccRCC.

The nine component genes identified to consist of long non-coding RNA (lncRNA) LINC01089, lncRNA LOC100128288, AI894139 pseudogene LOC155060, hect domain and RLD 2 pseudogene 2 HERC2P2, a non-protein-coding RNA SNHG10, and four protein-coding genes (SPACA6, RecQL4, ATXN7L2, and THSD7A). LncRNAs are known to regulate gene expression in part via their sponge actions towards miRNAs [27,28], and microRNAs commonly regulate multiple targets [29]. Importantly, non-coding RNAs contribute to renal cell carcinoma [30]. The evidence thus supports SigIQGAP1NW affecting multiple signaling events or processes, which might underlie its effectiveness in predicting poor prognosis of ccRCC.

LINC01089 has been reported in a limited number of articles (*n* = 2, PubMed) for a negative association with breast cancer metastasis in part via inhibition of the Wnt/β-catenin signaling [31,32]. SNHG10 enhances hepatocarcinogenesis and metastasis [33]. HERC2P2 was recently identified as a component gene in a 10-gene panel of blood transcripts that classifies the risk of breast cancer [34]. RecQL4 is one of five human RecQ helicases, with others being RecQ1, WRN, BLM, and RecQ5. Mutations in WRN, BLM, and RecQL4 cause Werner syndrome (WS), Bloom syndrome (BS), and Rothmud-Thomson syndrome (RTS), respectively, which are associated with premature aging, cancer predisposition, and chromosome abnormalities [35]. Elevations in RecQL4 display oncogenic activities in prostate cancer [36] and promote chemoresistance in gastric cancer [37]; evidence thus supports the important roles of RecQL4 in promoting tumorigenesis.

However, none of the nine component genes of SigIQGAP1NW has been reported in ccRCC (Table 5). Besides a modest significant association of THSD7A with poor OS, the rest of the component genes all robustly predict mortality risk evident by their *p* values (Table 4). Consistent with these observations, LINC01089, SPACA6, HERC2P2 (Figure 7) and others (Appendix A) effectively stratify ccRCCs with elevated fatality risk from those with a low fatality risk as individual genes. Furthermore, among four mRNA clusters of ccRCCs which resulted from unsupervised clustering of global mRNA expression [14], analyses using the recently established GEPIA2 dataset [24] reveal a significant elevation of HERC2P2 in mRNA cluster 2 tumors and significant downregulations of THSD7A expression in all mRNA clusters of ccRCC in comparison to the matched non-tumor controls (Figure 8). Taken together, evidence supports SigIQGAP1NW as a novel multigene panel with a potent biomarker potential in predicting poor prognosis of ccRCC.

## 3. Discussion

Nephrectomy remains the main curative treatment for patients with primary ccRCC; however, 20–40% of patients will develop recurrent disease [4], leading to metastasis. It is thus a critical task to classify tumors with a high- and low-risk of disease recurrence, metastasis, and poor prognosis to improve patient management. To meet this challenge, effective biomarkers are required.

We approach this clinical need through the investigation of a novel ccRCC factor IQGAP1. We demonstrate for the first time that downregulation of IQGAP1 is significantly associated with reductions of OS in ccRCC patients. The DEGs (*n* = 611) relative to IQGAP1 downregulation affect pathways regulating the typical functions of IQGAP1 [16,38], i.e., cytoskeleton involved cellular processes, including cellular component movement, chemotaxis, focal adhesion, response to mechanical stimulus, regulation of cell adhesion, extracellular extravasation, and cell matrix adhesion (Figure 1A). Furthermore, these DEGs affect important oncogenic pathways, including downregulation of immune responses, as well as upregulations of mitochondria-related oxidative phosphorylation and DNA damage repair (Table 1), all in line with ccRCC being a metabolic disease [39,40]. Downregulations of the immune system and abnormalities in DNA damage repair are major contributors to oncogenesis and cancer progression [41,42]. These observations thus suggest that IQGAP1 suppresses ccRCC, which is supported by the extensive conservation between IQGAP1 and the IQGAP2 tumor-suppressor [17,18,43,44]. However, IQGAP1 likely affects tumorigenesis in a complex manner. In most studies reported, IQGAP1 promotes oncogenesis [17]. As a scaffold protein, IQGAP1 facilitates the actions of RAF-MEK-ERK in part via binding ERK1/2 through its WW motif, contributing to ERK activation [45]. This property of IQGAP1 is at least essential in skin cancer, as targeting IQGAP1-facilitated ERK activation inhibited skin cancer tumorigenesis [45]. On the other hand, IQGAP1 displays tumor-suppressive functions in bladder cancer [19], a theme that is supported by our analyses of another urinary cancer ccRCC.

To further investigate the relevance of IQGAP1-associated network alterations (DEGs), we constructed a novel and robust multigene panel (SigIQGAP1NW) in assessing poor OS of ccRCC. We took advantage of two approaches in the signature construction: Random division of cohort into a training and testing population and the involvement of cross-validation in covariate selections from the training group. Although cross-validation can be equivalent to the conventional validation by splitting a dataset into a training set and a validation (testing) set, we trust that the inclusion of both in our study contributed to producing a robust gene signature. Additional favorable factors include the TCGA PanCancer Atlas ccRCC dataset being an excellent resource supporting OS biomarker studies [21].

An attractive feature of SigIQGAP1NW is its novelty, in which none of the nine component genes has been previously reported in ccRCC (Table 5). The importance of these genes, nonetheless, is validated by their effectiveness in stratifications of poor OS individually (Table 4; Figure 7; Appendix A) and their abilities to independently predict poor OS (except THSD7A) after adjusting for a set of clinical features and Winter Hypoxia Score. These properties outline their appealing clinical applications individually and as component genes of SigIQGAP1NW.

Another feature of SigIQGAP1NW is the inclusion of five non-protein-coding genes among its nine component genes (Table 5). With current knowledge of non-coding RNA being important in the regulation of networks rather than specific genes [46], SigIQGAP1NW likely affects complex networks, a potential underlying reason for the impressive robustness observed in this multigene panel in assessing poor OS. This is consistent with the current consensus for the importance of a multigene panel to possess multiple features or affecting multiple processes for it to be clinically useful in patient management [47]. Along this line, our first demonstration of a significant upregulation of the HERC2P2 pseudogene in mRNA cluster 2 (Figure 8), which is a more aggressive sub-type of ccRCCs than tumors in mRNA cluster 1 [14], further supports this research being novel and relevant. However, the limitations of this research include the lack of knowledge regarding the major networks or pathways that are affected by these non-coding RNAs; future research should investigate this aspect.

Among the four protein-coding component genes, they likely influence different aspects of oncogenesis. SPACA6 (Sperm Acrosome Associated 6) is an oocyte factor and contributes to sperm-egg fusion [48], with its oncogenic involvement unknown (Table 5). A similar situation applies to ATXN7L2; there is only one publication in PubMed suggesting its biomarker potential in non-small cell lung cancer [49]. RECQL4 is well known for its impact in maintaining genome stability and its involvement in tumorigenesis [50]. While its contribution to ccRCC has yet to be reported, RECQL4 is clearly important. Its functionality in genome stability is consistent with frequently mutations of a set of tumor-suppressor genes in ccRCC, including VHL, PBRM1, SETD1, and BAP1 [51]. Among 611 DEGs, RECQL4 is one of nine being selected for impact on OS, and importantly, it potently predicts the fatality risk of ccRCC (Appendix A). With respect to THSD7A, there are no reports for its oncogenic functions (Table 5). However, this protein is well-known for its impact on a kidney disease, membranous nephropathy [52]. Its inclusion in SigIQGAP1NW among 611 genes supports its potential impact on ccRCC, which agrees with its role in affecting kidney physiology. Furthermore, we provide a novel and additional support for its involvement in ccRCC, i.e., the across downregulation of THSD7A in all four mRNA clusters of ccRCC (Figure 8). Nonetheless, the protein expression status of these four protein-coding genes in ccRCC remains unknown, which needs to be determined in the context of ccRCC tumorigenesis and progression in the future.

Taken together, we have constructed a novel and robust multigene panel SigIQGAP1NW in predicting poor prognosis in ccRCC. While its potential in clinical applications is appealing, further investigations into SigIQGAP1NW potential in ccRCC need to be performed both retrospectively and prospectively. The latter aspect is likely more critical. Assessment of RCC recurrence is a major event of patient management; clinically, this risk has been evaluated using a set of clinical features, including stage, grade, particularly the TNM system, and others [53]. Eight clinical models have been developed, which explored a variety of clinical features, and are widely used in assessing the risk of RCC relapse, including Katten [54], Yaycioglu [55], UISS (University of California at Los Angeles Integrated Staging System) [56], SSIGN (Stage, Size, Grade, and Necrosis score) [57], Cindolo [58], Leibovich [59], MSKCC (Memorial Sloan Kettering Cancer Center) [60], and Karakiewicz [61]. A recent prospective validation of these models using patients with organ-confined high-grade disease (*n* = 1647) from the ASSURE cohort (Adjuvant Sorafenib or Sunitinib for Unfavorable Renal Carcinoma) revealed their modest prediction effectiveness with SSIGN being the best performer (C-index 0.688) which modestly outperformed the 2002 TNM system (C-index 0.603) [53]. Both UISS (C-index 0.556) and Yaycioglu (C-index 0.587) were underperformed compared to the 2002 TNM system [53]. While it may be informative to compare our SigIQGAP1NW to these clinical models for its predictive efficiency of ccRCC relapse, the TCGA PanCancer Atlas cohort does not support evaluations of these clinical models owing to the incomplete annotation of the relevant clinical features; for instance, Fuhrman grade, tumor size, and tumor necrosis, which are common features of these models [53], were not available in the TCGA PanCancer cohort. Nonetheless, it is appealing to examine whether SigIQGAP1NW will significantly improve the assessment of ccRCC relapse risk in combination with these clinical models. This possibility is supported by SigIQGAP1NW’s ability to predict PFS following resection of primary ccRCCs.

The modest performance of these widely used clinical models clearly calls for reliable molecular events-based biomarkers. Together with our report here, the search for ccRCC biomarkers has shown promising progress lately. Based on the metabolic feature of ccRCC, a 10-gene set was recently reported, which predicts ccRCC OS [62]. A set of 10 mRNAs and a group of 10 lncRNAs have been demonstrated to significantly assess OS possibility [63]. A multigene (*n* = 7) panel was very recently constructed with a property to significantly predict ccRCC OS [64]. Finally, groups of miRNAs with a relationship to long-term and poor OS have recently been reported [65]. While there does not appear to be an apparent connection between our SigIQGAP1NW and these biomarker sets in terms of component gene overlap, it is possible that related pathways might be commonly modeled. It will be intriguing to investigate whether potential combinations of SigIQGAP1NW with these will strengthen the predictive power. However, this possibility will take time and effort to be looked into in the future.

## 4. Materials and Methods

### 4.1. Patient Populations

cBioPortal [22,23] (https://www.cbioportal.org/) contains the most well-organized and comprehensive cancer genetic data for different cancer types. The TCGA PanCancer Atlas ccRCC dataset consists of 512 patients with primary ccRCC. All tumors have been surgically removed and profiled for RNA expression using RNA sequencing. The dataset has been well-demonstrated for its suitability in ccRCC OS biomarker studies [21].

### 4.2. Pathway Enrichment Analysis

Enrichment analyses were performed using Metascape [25] (https://metascape.org/gp/index.html#/main/step1) and Galaxy (https://usegalaxy.org/) for gene set enrichment.

### 4.3. Regression Analyses

Logistic regression was carried out with R. Cox proportional hazards (Cox PH) regression analyses were performed using the R *survival* package. The PH assumption was tested.

### 4.4. Construction of Multigene Signatures

The TCGA PanCancer Atlas ccRCC dataset within the cBioPortal database [22,23] (https://www.cbioportal.org/) was used to derive DEGs relative to IQGAP1 downregulation (*n* = 611). The random split of the dataset into a training and testing at 6:4 was performed using R. DEGs were selected for impact on OS using Elastic-net logistic regression within the *glmnet* package in R with 10-fold cross-validation. The mixing parameter of α was used at 0.5. At α = 0, Elastic-net operates as Ridge regression, which shrinks the coefficients of correlated predictors without performing covariate selection; at α = 1, it runs as Lasso, which tends to select one covariate among a group of related variables; this will reduce signature’s biomarker potential. Six rounds of selection at the setting were performed, and all unique genes obtained were combined into the final multigene panel SigIQGAP1NW.

### 4.5. Assignment of Signature Scores to Patients/Tumors

All component genes were examined for an association with OS shortening using multivariate Cox PH regression with the R “survival” package. The signature scores for individual patients were given using Sum (coef_1_ × gene_1exp_ + coef_2_ × gene_2exp_ + … …+ coef_n_ × gene_nexp_), where coef_1_ … coef_n_ are the coefs of individual genes and gene_1exp_ … … gene_nexp_ are the expression of individual genes.

### 4.6. Cutoff Point Estimation

Cutoff point to separate tumors with a high risk of mortality was estimated using Maximally Selected Rank Statistics (the *Maxstat* package) in R.

### 4.7. Odds Ratio Determination

OR analysis was performed using the *glmnet* package in R.

### 4.8. Statistical Analysis

Kaplan-Meier surviving curves and log-rank test were carried out using the R *survival* package, and tools provided by cBioPortal. Univariate and multivariate Cox regression analyses were run using the R *survival* package. Time-dependent receiver operating characteristic (tROC) analysis was performed using the R *timeROC* package. A value of *p* < 0.05 is considered statistically significant.

## 5. Conclusions

We provide the first evidence for a significant association of IQGAP1 reductions with ccRCC mortality. IQGAP1 downregulation correlates with network changes consisting of 611 DEGs; these DEGs are enriched in pathways important to ccRCC and the typical features of the disease. Furthermore, from the IQGAP1-associated network, a novel 9-gene signature (SigIQGAP1NW) has been constructed; it robustly predicts ccRCC fatality and recurrence after nephrectomy with a high level of certainty. Furthermore, all nine component genes of SigIQGAP1NW are novel to ccRCC, and 5/9 are novel to oncogenic functions in general. We also provide the first evidence for ccRCC-associated alterations in two component genes, HERC2P2 and THSD7A. This research may have a profound impact on ccRCC with respect to research and patient management.

## 6. Patents

A USA provisional patent for results reported in this manuscript was recently filed.

## Figures and Tables

**Figure 1 cancers-12-03471-f001:**
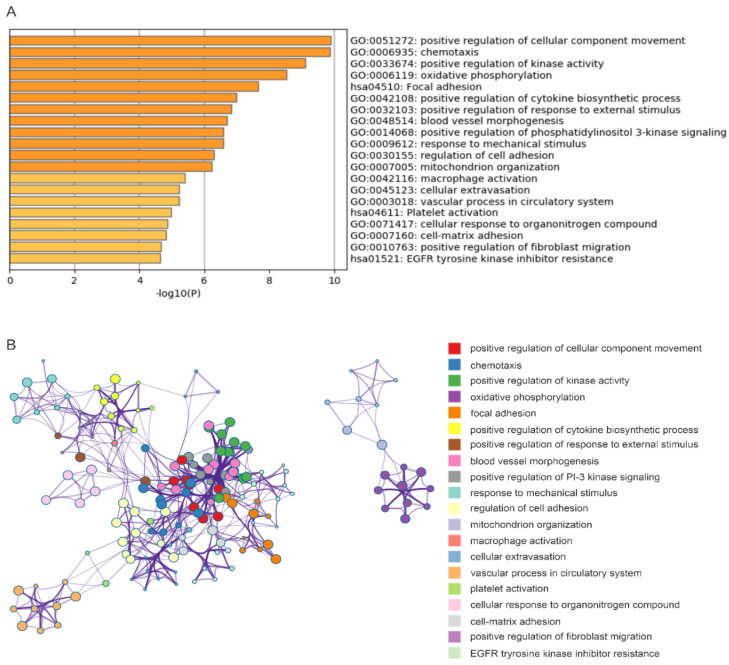
Pathway enrichment of DEGs relative to IQGAP1 downregulation. (**A**) Representatives of top clusters enriched. (**B**) Network presentation of those enriched clusters. Analyses were performed using Metascape [25].

**Figure 2 cancers-12-03471-f002:**
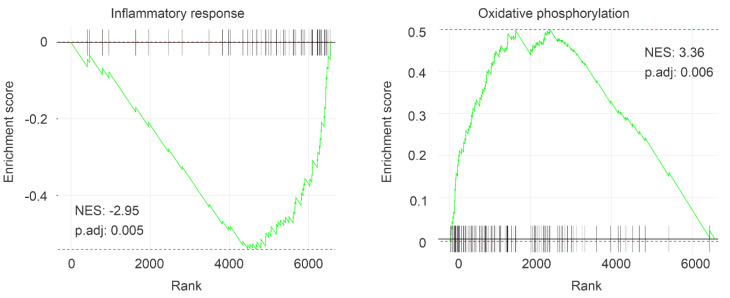
Gene set enrichment. IQGAP1 DEGs were analyzed for gene set enrichment among the human hallmark gene set. Two enriched gene sets functioning in the inflammatory response and oxidative phosphorylation are presented.

**Figure 3 cancers-12-03471-f003:**
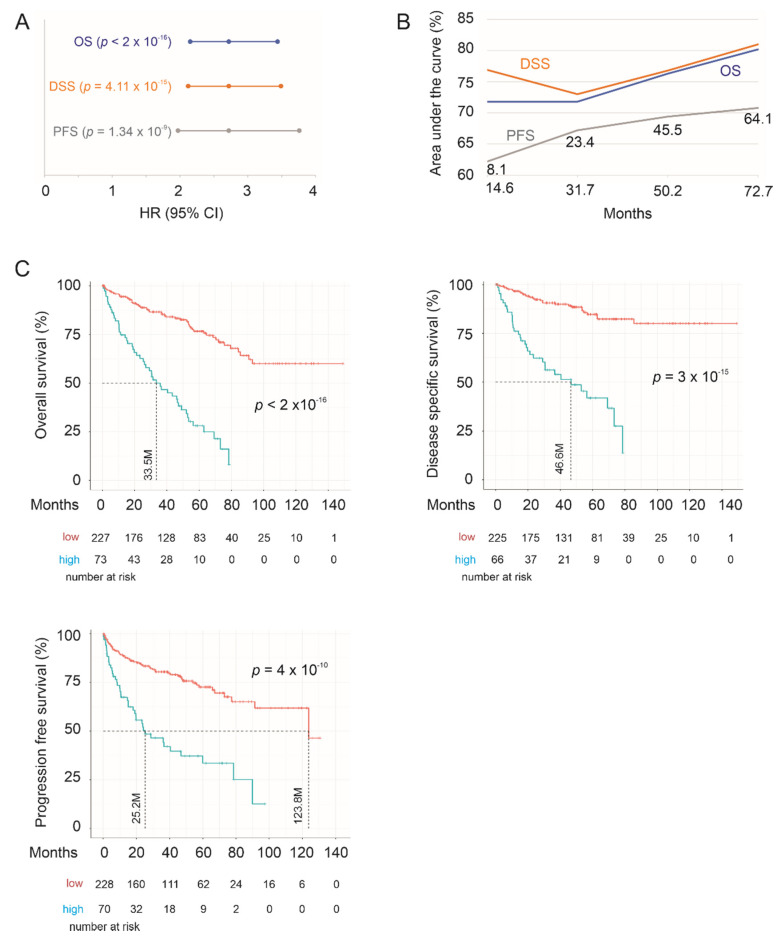
SigIQGAP1NW robustly stratifies risks of poor prognosis of ccRCC in the training sub-population. (**A**) HR, 95% CI, and *p* values. OS, overall survival; DSS, disease-specific survival; PFS, progression-free survival. (**B**) Time-dependent ROC (receiver operating characteristic) curve. Months for PFS are specifically labeled. (**C**) Kaplan Meier survival curves. Statistical analyses were performed using logrank test.

**Figure 4 cancers-12-03471-f004:**
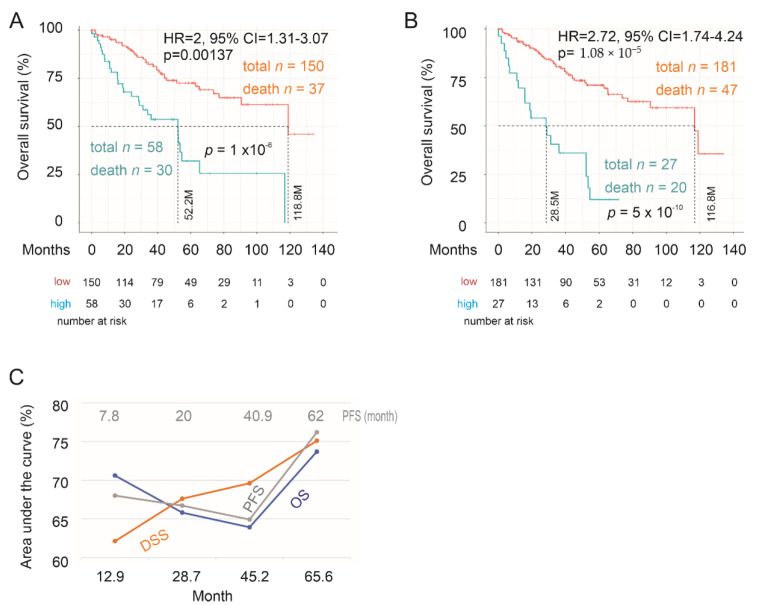
SigIQGAP1NW efficiently predicts the risks of poor prognosis of ccRCC in the testing population. (**A**,**B**) Analyses of the fatality risk of ccRCC using SigIQGAP1NW scores defined from the training population (**A**) or the testing cohort (**B**). HR, 95% CI, and *p* values along with logrank *p* value for Kaplan Meier survival curves are provided. The respective median survival times are also indicated. (**C**) Time-dependent ROC curve. Months for PFS are specifically labeled.

**Figure 5 cancers-12-03471-f005:**
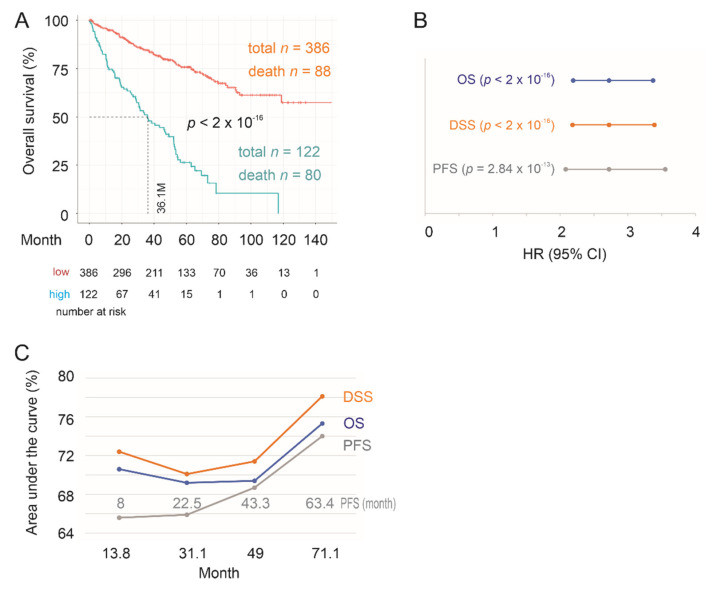
SigIQGAP1NW classifies the risks of poor prognosis of ccRCC in the TCGA PanCancer Atlas ccRCC cohort with a high degree of certainty. (**A**) Kaplan Meier survival curve. (**B**) HR, 95% CI, and *p* values for the indicated ccRCC events. (**C**) Time-dependent ROC curve. Months for PFS are specifically labeled.

**Figure 6 cancers-12-03471-f006:**
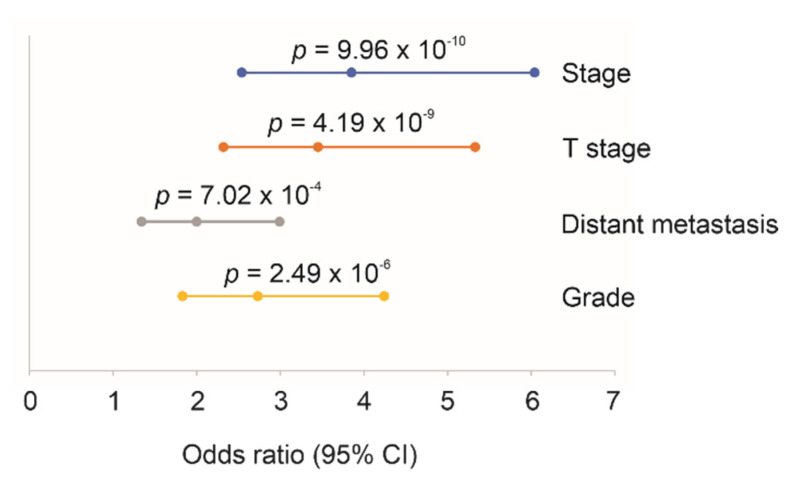
Association of SigIQGAP1NW with worse clinical features of ccRCC. Stage 1 and 2 are expressed as “0”, while Stage 2 and 4 are represented as “1”. T stages 1 and 2 are converted to “0”; T3 and T4 are combined to “1”. Grades 1 and 2 are used at “0”; and Grades 3 and 4 are converted to “1”. SigIQGAP1NW scores are used for analysis.

**Figure 7 cancers-12-03471-f007:**
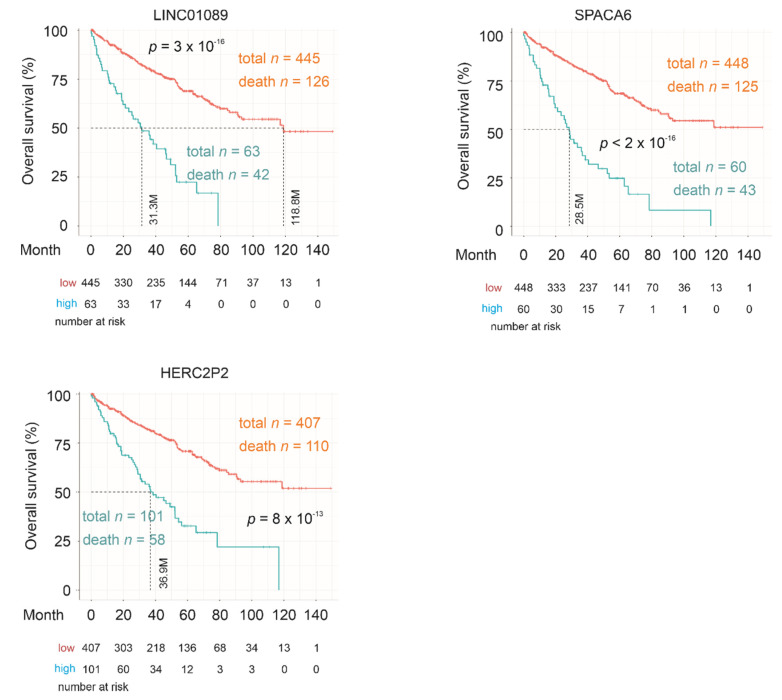
Association of SigIQGAP1NW component genes with poor OS of ccRCC. Kaplan Meier survival curves for the indicated component genes along with logrank *p* values, median survival month, and other information are included.

**Figure 8 cancers-12-03471-f008:**
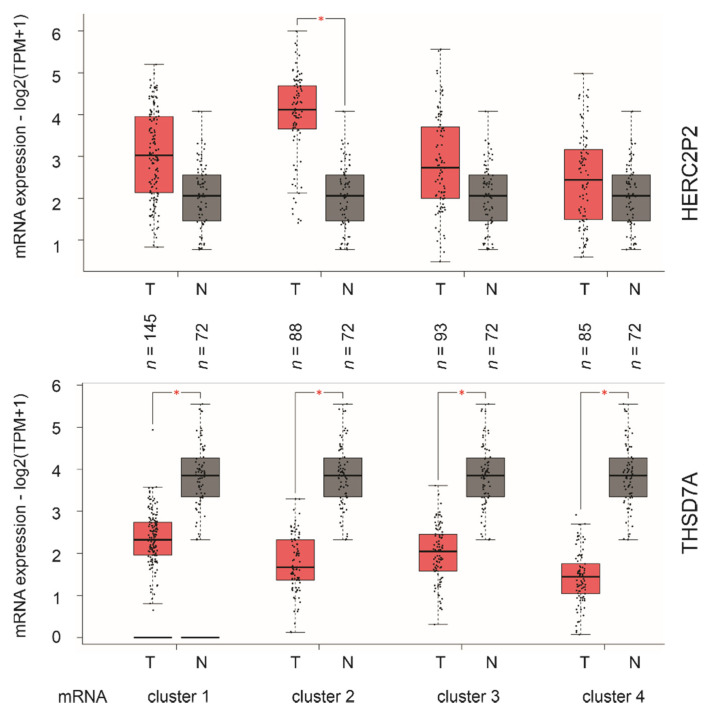
Differential expression of the HERC2P2 and THSD7A component genes in ccRCC (Tumor/T) and matched non-tumor kidney tissues (N). Gene expression was determined by RNA-seq (TCGA) and analyzed using the GEPIA2 program. Four mRNA clusters are indicated. TPM, transcripts per million. Statistical analyses were performed by GEPIA2, * *p* < 0.05.

**Table 1 cancers-12-03471-t001:** Human hallmark gene set enrichment of IQGAP1 DEGs.

Pathway	*P*.Adj	ES	NES	Size
INFγ response	0.005545	−0.37972	−2.2396	79
Il2_Stat5 signaling	0.005545	−0.39412	−2.3246	79
Mitotic spindle	0.005545	−0.36571	−2.3260	115
UV response DN (down)	0.005545	−0.4882	−2.8751	78
Allograft rejection	0.005545	−0.4118	−2.3087	67
TNFα signaling via NFκB	0.005545	−0.48317	−2.5962	57
Estrogen response early	0.005545	−0.39141	−2.1554	63
Complement	0.005545	−0.36543	−2.1273	76
Apical junction	0.005545	−0.41499	−2.3820	73
Epithelial mesenchymal transition	0.005545	−0.56703	−3.0	62
KRAS signaling UP	0.005545	−0.60277	−3.2622	59
Inflammatory response	0.005545	−0.54082	−2.9457	61
TGFβ signaling	0.005545	−0.50114	−2.1570	26
Androgen response	0.005545	−0.44591	−2.2861	50
Coagulation	0.005545	−0.46409	−2.1825	37
Oxidative phosphorylation	0.006152	0.50062	3.3574	118
DNA repair	0.011239	0.29602	1.8111	83
Adipogenesis	0.011239	0.28891	1.8073	91
IL6 Jak Stat3 signaling	0.012913	−0.43411	−1.9242	29
Fatty acid metabolism	0.028448	0.31956	1.7648	52
G2M checkpoin	0.033794	−0.27572	−1.6313	81
Spermatogenesis	0.033962	−0.38359	−1.7162	31
Protein secretion	0.048784	−0.28187	−1.5703	66

*p*.adj, adjusted *p* value; ES, enrichment score; NES, normalized enrichment score.

**Table 2 cancers-12-03471-t002:** Composition of SigIQGAP1NW.

Gene	Locus	Log2 Ratio ^1^	*p*-Value	*q*-Value
LINC01089	12q24.31	1.19	4.84 × 10^−17^	3.69 × 10^−16^
SPACA6	19q13.41	1.12	2 × 10^−16^	1.41 × 10^−15^
LOC155060	7q36.1	1.09	5.01 × 10^−11^	1.90 × 10^−10^
LOC100128288	17p13.1	1.08	3.54 × 10^−19^	3.72 × 10^−18^
SNHG10	14q32.13	1.05	8.40 × 10^−17^	6.19 × 10^−16^
RECQL4	8q24.3	1.01	2.67 × 10^−17^	2.11 × 10^−16^
HERC2P2	15q11.2	0.97	2.52 × 10^−8^	7.14 × 10^−8^
ATXN7L2	1p13.3	0.95	3.15 × 10^−14^	1.69 × 10^−13^
THSD7A	7p21.3	−1.44	3.38 × 10^−14^	1.81 × 10^−13^

^1^ ccRCCs with IQGAP1 downregulation compared to those without the downregulation.

**Table 3 cancers-12-03471-t003:** Univariate and multivariate Cox analysis of SigIQGAP1NW for poor OS of ccRCC.

Factors	Univariate Cox Analysis			Multivariate Cox Analysis		
	HR	95% CI	*p*-Value	HR	95% CI	*p*-Value
Sig ^1^	2.72	2.19–3.37	<2 × 10^−16^ ***	1.80	1.41–2.31	2.85 × 10^−6^ ***
Age ^2^	1.03	1.02–1.04	2.78 × 10^−6^ ***	1.03	1.02–1.05	6.34 × 10^−5^ ***
Sex ^3^	0.96	0.70–1.31	0.793	1.08	0.77–1.51	0.64719
State III ^4^	2.8	1.84–4.24	1.28 × 10^−6^ ***	1.82	1.18–2.85	0.00714 **
State IV ^4^	6.83	4.60–10.12	<2 × 10^−16^ ***	4.49	2.85–7.07	9.30 × 10^−11^ ***
Grade 3 ^5^	1.94	1.32–2.86	0.00075 ***	1.22	0.81–1.84	0.34185
Grade 4 ^5^	5.38	3.59–8.05	3.06 × 10^−16^ ***	1.73	1.07–2.81	0.02540 *
WHS ^6^	1.04	1.03–1.05	2.19 × 10^−11^ ***	1.01	1.002–1.03	0.02525 *

^1^ SigIQGAP1NW score; ^2^ Age at diagnosis; ^3^ Compared to females; ^4^ Compared to Stage I, Stage II is not significant at univariate Cox analysis; ^5^ Compared to Grade 1 + 2 (both grades were combined because of small sample number for Grade 1 samples, *n* = 12); ^6^ Winter hypoxia score; *, **, *** for *p* < 0.05, 0.01, and 0.001, respectively.

**Table 4 cancers-12-03471-t004:** Association of SigIQGAP1NW component genes with poor OS of ccRCC.

Gene	HR	95% CI	*p* Value
LINC01089	1.002	1.001–1.002	3.14 × 10^−13^ ***
SPACA6	1.006	1.005–1.008	2.69 × 10^−15^ ***
LOC155060	1.002	1.001–1.003	1.31 × 10^−13^ ***
LOC100128288	1.008	1.005–1.01	4.53 × 10^−8^ ***
SNHG10	1.007	1.005–1.009	5.9 × 10^−10^ ***
RECQL4	1.003	1.002–1.004	9.7 × 10^−14^ ***
HERC2P2	1.0	1.0–1.001	5.64 × 10^−11^ ***
ATXN7L2	1.005	1.004–1.006	5.24 × 10^−14^ ***
THSD7A	0.9991	0.9984–0.9999	0.0292 *

* *p* < 0.05; *** *p* < 0.0001.

**Table 5 cancers-12-03471-t005:** Oncogenic functions of SigIQGAP1NW component genes.

Gene	Gene Details	ccRCC ^1^	Oncogenesis ^2^	Refs
LINC01089	long intergenic non-protein-coding RNA 1089	unknown	inhibition of breast cancer metastasis	[32]
SPACA6	sperm acrosome associated 6	unknown	unknown	NA
LOC155060	AI894139 pseudogene	unknown	unknown	NA
LOC100128288	uncharacterized, an RNA gene that is affiliated with the lncRNA class	unknown	unknown	NA
SNHG10	small nucleolar RNA host gene 10, a non-protein-coding RNA	unknown	promotion of hepatocellular carcinoma metastasis via activating c-Myb	[33]
RECQL4	RecQ like helicase 4	unknown	Driving gastric cancer resistance to cisplatin via activating the AKT-YB1-MDR pathway	[37]
HERC2P2	hect domain and RLD 2 pseudogene 2	unknown	A component of a 10-gene blood biomarker of breast cancer	[34]
ATXN7L2	ataxin 7 like 2	unknown	Not clear	NA
THSD7A	thrombospondin type 1 domain containing 7A	unknown	Not clear	NA

^1^ potential functioning in ccRCC; ^2^ function in tumorigenesis.

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
