# Peer review of "Construction of a Novel Multigene Panel Potently Predicting Poor Prognosis in Patients with Clear Cell Renal Cell Carcinoma"

_cancers, 2020, doi:10.3390/cancers12113471_

Round 1

Reviewer 1 Report

Thank you for the opportunity to review the manuscript entitled, "Construction of a novel multigene panel potently predicting poor prognosis in patients with clear cell renal cell carcinoma." The authors present analysis of a novel gene signature of RCC, generated using comparisons between IQGAP1 expressing and IQGAP1 down-regulated RCCs. The use of IQGAP1 in this analysis is interesting. Previous reports have studied IQGAP1 in cancer for its potential role in scaffold-kinase signal transduction, and in particular in RAS mutant tumors. As RCC is not a RAS mutant tumor, but is known to have significant MAPK signaling and respond to tyrosine kinase inhibitor therapy. The manuscript identifies 9 genes in this panel that are associated with overall survival using TCGA data. The research is interesting and addresses an important clinical need. However, I do have several suggestions to improve the manuscript. 

Introduction: The introduction does suggest that the authors are not as familiar with the current clinical management of RCC. For example, the authors could include a reference for kidney cancer being the 9th and 14th most common cancer. Perhaps this is international data, or data from Canada specifically? It is more common in than this in the USA for example. The authors also suggest that 30-40% of patients with localized RCC develop metastasis after surgery. This is clearly outdated and far from the current state of RCC care. The vast majority of patients have no recurrence of disease in the 5 years after surgery. The authors also state that mRCC is incurable, which is problematic as there are known durable responses to immunotherapy based therapies. The introduction also misrepresents the current 1st line treatments for patients with mRCC, which typically include checkpoint inhibitor therapies (alone and in combination with tyrosine kinase inhibitors).  The introduction also omits a landmark IQGAP1 study (PMID 23603816). This reference, and other similar articles, suggest that blocking IQGAP1 faciliated signaling may be a cancer treatment strategy. This is important to consider as this article suggests that lower IQGAP1 expression is protective in RCC.

The manuscript generates this 9 gene score and correlates this with clinical features at presentation in using TCGA data. There are several things that should be revised in these analyses. First, the gene expression of IQGAP1, and the panel, should be correlated with grade. Second, T stage is not a direct surrogate for tumor size. While it is true that there are size cutoffs for some T stages, other T stages are defined by invasion into perirenal fat regardless of size (T3a), large vessels (T3), or local invasion into adjacent organs (T4). These differences become important when building multivariable models to determine if the gene signature is an independent predictor. These multivariable models should not include overall stage along with individual T and M staging, as these are collinear and could result in model instability. That is, overall stage is based on the T N and M stages, and can not be included in the model along with these individual factors. Second these models should include tumor grade, which is an established predictor of outcomes. These models should also include patient sex. It is not clearly justified why the Winter hypoxia score would be included in these models. 

The limitations of this analysis should be clearly stated. These include the lack of correlation with protein levels of these genes and lack of validation outside of the TCGA. More discussion is needed to explain how this may fit in a previous model that suggested interfering with IQGAP1 signaling would be a treatment strategy in other tumors. 

Author Response

We appreciate the reviewer’s insightful remarks. Here are our detailed revisions.

Reviewer #1 - “Introduction: The introduction does suggest that the authors are not as familiar with the current clinical management of RCC. For example, the authors could include a reference for kidney cancer being the 9th and 14th most common cancer. Perhaps this is international data, or data from Canada specifically? It is more common in than this in the USA for example. The authors also suggest that 30-40% of patients with localized RCC develop metastasis after surgery. This is clearly outdated and far from the current state of RCC care. The vast majority of patients have no recurrence of disease in the 5 years after surgery. The authors also state that mRCC is incurable, which is problematic as there are known durable responses to immunotherapy based therapies. The introduction also misrepresents the current 1st line treatments for patients with mRCC, which typically include checkpoint inhibitor therapies (alone and in combination with tyrosine kinase inhibitors).  The introduction also omits a landmark IQGAP1 study (PMID 23603816). This reference, and other similar articles, suggest that blocking IQGAP1 faciliated signaling may be a cancer treatment strategy. This is important to consider as this article suggests that lower IQGAP1 expression is protective in RCC.”

Authors' response – With respect to the tumor incidence, our statement for the 9th and 14th rank of kidney cancer in men and women respectively was based on international data, as Reviewer #1 has suggested. References to this statement have been added. Kidney cancer is more prevalent in the US, which has also been included with corresponding reference (lines 45-48). We thank the reviewer for this precise comment.

     As reviewer has pointed out, we have overstated the rate of ccRCC metastasis, which is approximately 5-15% following initial resection. In this revision, we have corrected our statements in both Introduction and Discussion (marked with red). Recurrence rate is 20-40% for primary ccRCC after surgery, which is in line with our initial statement. However, we agree with the reviewer that not every recurrence occurs in the first 5 years; to reflect this fact, we included the word “eventually” to describe the recurrence (line 52); references have been cited.

     We thank Reviewer #1 for pointing out the new first-line therapy in treating metastatic ccRCC. This update has been included (lines 53-57, marked with red). Even with the new first-line therapy, resistance remains a challenge as majority of patient will still progress; proper citation for this knowledge is provided. Nonetheless, we see the reviewer’s point regarding whether ccRCCs are currently incurable and have deleted this statement.

     We appreciate Reviewer #1 for pointing out the 2013 Nat Med article supporting a critical role of IQGAP1 in facilitating ERK activation as a scaffold protein in skin cancer. The implication of this publication has been discussed (Discussion, lines 323-329).

Reviewer #1 - “The manuscript generates this 9 gene score and correlates this with clinical features at presentation in using TCGA data. There are several things that should be revised in these analyses. First, the gene expression of IQGAP1, and the panel, should be correlated with grade. Second, T stage is not a direct surrogate for tumor size. While it is true that there are size cutoffs for some T stages, other T stages are defined by invasion into perirenal fat regardless of size (T3a), large vessels (T3), or local invasion into adjacent organs (T4). These differences become important when building multivariable models to determine if the gene signature is an independent predictor. These multivariable models should not include overall stage along with individual T and M staging, as these are collinear and could result in model instability. That is, overall stage is based on the T N and M stages, and can not be included in the model along with these individual factors. Second these models should include tumor grade, which is an established predictor of outcomes. These models should also include patient sex. It is not clearly justified why the Winter hypoxia score would be included in these models.”

Authors' response – These insightful remarks are highly appreciated. The information that T stage is not equivalent to tumor size has been precisely stated in this revision (Fig 6 and others). We agree with the Reviewer regarding TNM staging-based overall stage or staging groups. T and M stages have been removed with grade added in multivariate analyses (Table 3), as reviewer has suggested. All related HR, 95% CI, and p values have been corrected throughout the manuscript. However, these modifications do not affect the conclusions.

Reviewer #1 - “The limitations of this analysis should be clearly stated. These include the lack of correlation with protein levels of these genes and lack of validation outside of the TCGA. More discussion is needed to explain how this may fit in a previous model that suggested interfering with IQGAP1 signaling would be a treatment strategy in other tumors”

Authors' response – Limitations as pointed by Reviewer #1 have been added (lines 354-356 and lines 371-373). Potential applications of our signature to the existing widely used 8 clinical models are discussed (see the new paragraph, lines 374-396).

Reviewer 2 Report

The authors use TCGA Pan cancer RCC atlas gene expression analysis of genes that associate with IQGAP1 regulation. Previous even some large analyses have not suggested any of the reported 9 genes as prediction markers. While the 9 gene signature might have some potential for prediction of overall ccRCC survival, the analysis as it stands now won`t be clinically useful.

My main criticism is that the patient material has not been described in the text nor taken into account in the analysis.

- Has the analysis been done from primarily metastatic patients (M1) or patients undergone (partial) nephrectomy without detectable metastasis (M0) at the time of operation or both as is suggested by Table S2? My suggestion is that these two situations are separated and the PFS/OS analysis is done for both separately. Mixing all RCC patients into one pool will make any prediction model useless for clinicians and undermines the title and the second sentence of discussion.

- This is also because most M1 patients have probably received TKI or other medication at some point even though some might have been under surveillance initially. That would obviously affect OS and in particular PFS. How was this taken into account when randomising into testing and training population?

- Second, for clinical use there are several prediction models. Separately predicting apperance of metastasis / PFS for M0 and OS for M1 patients. Models for M1 include the IMDC score, and for M0 situation for example UISS and Karakiewicz models for all RCC types and the SSIGN, Leibovich (2003) and Sorbellini prediction models for ccRCC. Since these are widely used it would be of great interest to see how the proposed prediction signature compares with UISS, SSIGN or Leibovich. I am confident that the TCGA atlas provides sufficient information to perform a comparison against one of the approved clinical models.

Author Response

We thank the reviewer for his/her comments on the clinical aspect. The reviewer’s remarks focused on two areas: cohort details regarding the ccRCC origin (primary vs metastatic ccRCC) and the relationship of our panel to the existing clinical models. Please see our detailed responses.

Reviewer #2 - “My main criticism is that the patient material has not been described in the text nor taken into account in the analysis.

- Has the analysis been done from primarily metastatic patients (M1) or patients undergone (partial) nephrectomy without detectable metastasis (M0) at the time of operation or both as is suggested by Table S2? My suggestion is that these two situations are separated and the PFS/OS analysis is done for both separately. Mixing all RCC patients into one pool will make any prediction model useless for clinicians and undermines the title and the second sentence of discussion.

- This is also because most M1 patients have probably received TKI or other medication at some point even though some might have been under surveillance initially. That would obviously affect OS and in particular PFS. How was this taken into account when randomising into testing and training population?”

Authors' response – We thank the reviewer for these critical comments regarding whether tumors in the TCGA PanCancer cohort were organ-confined or metastatic. We apologize for not clearly stating the primary nature of these ccRCCs, i.e. all ccRCC included in the TCGA PanCancer cohort were primary; this has been clearly stated in this revision (lines 154-160 and lines 230-237) with reference cited. With this issue clarified, we wonder if separate analyses of SigIQGAP1NW in M0 and M1 (as there are no true M1 tumors being profiled for gene expression in this cohort) population are necessary. Nevertheless, the primary ccRCCs that are associated with metastasis are high risk tumors. With this in mind, we analyzed SigIQGAP1NW value in predicting poor OS in M0 and the primary ccRCC with confirmed metastasis development (“M1”); our 9-gene panel significantly predicts poor OS in both groups. The data and its proper interpretation are included (lines 230-237).

Reviewer #2 - “- Second, for clinical use there are several prediction models. Separately predicting apperance of metastasis / PFS for M0 and OS for M1 patients. Models for M1 include the IMDC score, and for M0 situation for example UISS and Karakiewicz models for all RCC types and the SSIGN, Leibovich (2003) and Sorbellini prediction models for ccRCC. Since these are widely used it would be of great interest to see how the proposed prediction signature compares with UISS, SSIGN or Leibovich. I am confident that the TCGA atlas provides sufficient information to perform a comparison against one of the approved clinical models.”

Authors' response – We see the significance of these insightful comments. For the remarks related to IMDC (International Metastatic RCC Database Consortium) risk score, the primary nature of ccRCCs in the TCGA PanCancer cohort will not support this analysis. On the other hand, the relevant clinical events were incomplete to support this analysis. 1) The neutrophile feature was not available; 2) other clinical characters were fragmented to very high levels which likely makes analysis not very meaningful; for instance, there were n=3 patients with KARNOFSKY_PERFORMANCE_SCORE < 80% and n=10 patients with elevated Serum_Calcium_level.

     The suggested comparisons of SigIQGAP1 (our gene panel) with the widely used clinical models (UISS, Karakiewicz, SSIGN, Leibovich, and Sorbellini/MSKCC) are insightful comments. However, the incomplete annotation of clinical features in TCGA PanCancer ccRCC cohort does not support a comparison with any of these 5 clinical models and the other 3 clinical models (Kattan, Yaycioglu, and Cindolo). For example, “fuhrman nuclear grade”, “tumor size”, “necrosis”, and “symptoms at presentation” (the feature required for these analyses) are not present. Nonetheless, we discussed these clinical models, the reasons why comparison would be unsuitable in this case, and the need to examine these comparisons in the future (lines 374-396). Although such comparisons are not feasible due to technical issues, the addition of this content in this revision clearly enhances this manuscript, to which we are thankful to Reviewer #2’s remarks.

Reviewer 3 Report

The authors describe a new panel composed of 9 genes with predictive potential of bad prognosis in clear cell renal cell carcinoma. Although the authors discuss in deep their results, they do not comment on other previously published attempts to achieve a genomic signature of prognosis in CCRCC. In this sense, the authors should include at least a paragraph in the discussion reviewing, commenting and comparing other published results. There are many examples of the same attempt only in 2020, for example: Sci Rep 2020, 10: 12949, Oncol Lett 2020; 20: 2420, Curr Med Sci 2020; 40: 773, Hereditas 2020; 157: 38, PLoS One 2020; 15: e0238809

Author Response

Reviewer #3 - “The authors describe a new panel composed of 9 genes with predictive potential of bad prognosis in clear cell renal cell carcinoma. Although the authors discuss in deep their results, they do not comment on other previously published attempts to achieve a genomic signature of prognosis in CCRCC. In this sense, the authors should include at least a paragraph in the discussion reviewing, commenting and comparing other published results. There are many examples of the same attempt only in 2020, for example: Sci Rep 2020, 10: 12949, Oncol Lett 2020; 20: 2420, Curr Med Sci 2020; 40: 773, Hereditas 2020; 157: 38, PLoS One 2020; 15: e0238809”

Authors' response – We are grateful to the reviewer’s effort with updates of similar research efforts that are recently published. As the reviewer has indicated, biomarker research in ccRCC has been progressing in recent years. We are glad to see these efforts and have discussed these publications and their potential connections to our gene panel (the last paragraph or lines 397-408).